Between tide and wave marks: a unifying model of physical zonation on littoral shores

Bird Christopher E. 1 2
Franklin Erik C. 2
Smith Celia M. 3
Toonen Robert J. 2 rjtoonen@gmail.com
1 Department of Life Sciences, Texas A&M University - Corpus Christi , Corpus Christi, TX , United States
2 Hawaiʻi Institute of Marine Biology, School of Ocean and Earth Science and Technology, University of Hawaiʻi , Kaneohe, HI , United States
3 Department of Botany, University of Hawaiʻi , Honolulu, HI , United States
Bruno John
Electronic publication date: 2013 Sep 19
Publication date: 2013
Volume: 1
Electronic Location ID: e154
Received 2013 Jul 16; Accepted 2013 Aug 18
Copyright: © 2013 Bird et al.
Copyright year: 2013
Copyright holder: Bird et al.
License: This is an open access article distributed under the terms of the Creative Commons Attribution License, which permits unrestricted use, distribution, and reproduction in any medium, provided the original author and source are credited.
License URL: https://creativecommons.org/licenses/by/3.0/

Keywords: Desiccation, Stress gradient, Temperature, Disturbance, Predictive model, Intertidal environment, Immersion, Emersion, Hawaiʻi, Intertidal

Funding: University of Hawaiʻi Ecology, Evolution, and Conservation Biology (EECB) program University of Hawaiʻi Sea Grant College Program Hawaiʻi Coral Reef Initiative NOAA ONMS MOA 2005-008/6882 NSF Bio OCE-1260169 This research was funded in part by a graduate research grant from the University of Hawaiʻi Ecology, Evolution, and Conservation Biology (EECB) program (to CEB), a project development award from the University of Hawaiʻi Sea Grant College Program (CMS & RJT), the Hawaiʻi Coral Reef Initiative (RJT & CEB), NOAA Office of National Marine Sanctuaries, MOA 2005-008/6882 (to RJT), and the National Science Foundation Grant OCE-1260169 (RJT). The funders had no role in study design, data collection and analysis, decision to publish, or preparation of the manuscript.

==============================
The effects of tides on littoral marine habitats are so ubiquitous that shorelines are commonly described as ‘intertidal’, whereas waves are considered a secondary factor that simply modifies the intertidal habitat. However mean significant wave height exceeds tidal range at many locations worldwide. Here we construct a simple sinusoidal model of coastal water level based on both tidal range and wave height. From the patterns of emergence and submergence predicted by the model, we derive four vertical shoreline benchmarks which bracket up to three novel, spatially distinct, and physically defined zones. The (1) emergent tidal zone is characterized by tidally driven emergence in air; the (2) wave zone is characterized by constant (not periodic) wave wash; and the (3) submergent tidal zone is characterized by tidally driven submergence. The decoupling of tidally driven emergence and submergence made possible by wave action is a critical prediction of the model. On wave-dominated shores (wave height ≫ tidal range), all three zones are predicted to exist separately, but on tide-dominated shores (tidal range ≫ wave height) the wave zone is absent and the emergent and submergent tidal zones overlap substantially, forming the traditional “intertidal zone”. We conclude by incorporating time and space in the model to illustrate variability in the physical conditions and zonation on littoral shores. The wave:tide physical zonation model is a unifying framework that can facilitate our understanding of physical conditions on littoral shores whether tropical or temperate, marine or lentic.

Introduction

Littoral habitats, those lying between the low-tide line and the upper limit of aquatic species on the shore, are among the most studied and well-known aquatic habitats. Much attention has been devoted to the study of organisms on rocky shores - in particular their vertical zonation, the upper and lower limits of species, and distribution along gradients of wave exposure. Hypotheses addressing the causes of biotic zonation and community structure have evolved from strictly physical (Colman, 1933) to an inseparable combination of physical and biological factors, including physiological tolerance (Connell, 1961a; Connell, 1961b; Somero, 2002) and species interactions (Bruno & Bertness, 2001; Menge & Branch, 2001). Throughout the progression of intertidal zonation research, the most widely accepted paradigm has remained that the predictable pattern of tidal rise and fall is the “primary” mechanism affecting shoreline water levels and the littoral habitat (Lewis, 1964; Ricketts et al., 1985; Stephenson & Stephenson, 1972).

Stephenson & Stephenson (1949) and Stephenson & Stephenson (1972) proposed their “universal features of zonation between tide-marks on rocky coasts” after conducting surveys of vertical biotic zonation on littoral shores world-wide. While the model of Stephenson & Stephenson (1972) focuses on biotic zonation, it is essentially a physical model based on the concept that there is a classic intertidal zone (midlittoral zone = balanoid zone), a transition zone between the intertidal zone and the terrestrial biosphere (supralittoral fringe = Littorina zone), and a transition zone between the intertidal zone and subtidal biosphere (infralittoral fringe = laminarian zone). One of the benefits of focusing on the biota, rather than elevation, was that these zones were not found predictably at the same height above sea level either within or among geographic regions. “Secondary modifying factors” of littoral climate, such as waves, were thought to enlarge and/or elevate the basic zones, but not change their primary properties (Lewis, 1964; Ricketts et al., 1985; Stephenson & Stephenson, 1972). More recent research continues to demonstrate that biotic zones and species distribution limits do not consistently occur at the same shore levels, even within shores (Benedetti-Cecchi & Cinelli, 1997).

A fundamental advance in the understanding of biotic zonation on rocky shores was the demonstration that species interactions also affected zonation patterns, where biotic factors generally determine the lower limit of distribution and physical factors affect the upper limit of distribution (Connell, 1961a; Connell, 1961b; Paine, 1974). A number of exceptions to this generalization have been demonstrated (Bertness, 1989; Bertness & Leonard, 1997; Bertness et al., 1999; Choat, 1977; McLay & McQueen, 1995; Robles & Desharnais, 2002; Wootton, 1992), many of which highlight the effect of biotic interactions on the realized distribution of a species. These examples demonstrate that biotic factors can also regulate the upper limits of a species’ distribution, but focus on the proximate factors affecting realized species distributions. Ultimately, the inseparable interaction between physical and biotic factors define the realized limits of species (Denny & Wethey, 2001), and models of physical clines can contribute valuable information to elucidate biotic processes (see Robles & Desharnais, 2002). Indeed, “rocky shores are the ‘stage’ upon which ecological ‘dramas’ are played out, and physical conditions both provide the ‘ambiance’ and help direct the ‘plays’ (pg 221, Menge & Branch, 2001).” Consequently, a more predictive model of physical habitat zonation on littoral shores would be very valuable.

Patterns of changing community structure and composition along wave exposure gradients have not typically been viewed as an issue of zonation. Instead, shifts in community structure along wave exposure gradients have traditionally been associated with the effects of hydrodynamic force (physical stress and disturbance) on biotic interactions (Menge, 1978; Menge & Sutherland, 1987). This is an oversimplification, however, because with increasing wave exposure factors other than hydrodynamic force are at play on shores where wave heights approach or exceed the tidal range. More recent studies demonstrate that waves also affect patterns of community composition along wave exposure gradients by creating habitat with associated physiological stress and disturbance that does not exist on more wave protected shores (Burrows, Harvey & Robb, 2008; Harley, 2003; Harley & Helmuth, 2003; Thomas, 1986). In particular, GIS-based models of wave exposure and quantitative metrics of effective fetch can explain a considerable proportion of variance in species abundances among sites at some locations (Burrows, Harvey & Robb, 2008; Thomas, 1986). Even so, the manner and mechanism by which waves interact with tides to create littoral habitat and the subsequent effects on biological processes have yet to be fully explored.

Here we investigate the roles of tides and waves in driving the characteristics of the physical habitat by deriving a sinusoidal model of coastal water level. We begin with a model of coastal water level based solely on tidal range and wave height. We use the model to derive relevant physically-defined shoreline benchmarks and partition the littoral zone into novel habitats. In so doing, we develop a system for the generic physical quantification and categorization of any shoreline across a range of spatial and temporal scales.

Materials and Methods

Model of coastal water level

We construct our simple model of coastal water level using both tidal and wave signals. First, we model the coastal water level (W) over time by summing sinusoidal models of water level due to tidal (WT) and wave action (WW) as follows: (1) WT=AT+ATsin2πTPT(Fig. 1A)

(2) WW=AWsin2πTPW/3600(Fig. 1B)

(3) W=WT+WW(Fig. 1C)

where AT is the tidal amplitude in meters, T is the time in number of hours, PT is the tidal period (PT = 12.2 h), AW is the wave amplitude, and PW is the wave period (PW = 10 s, note that the wave period has been increased in the figures for aesthetic reasons). The goal of this model is to grossly estimate the patterns of emersion and submersion experienced by shoreline organisms and demonstrate the conceptual consequences of taking wave height into account. Many additional factors interact with wave height to determine the extent of wave run-up and splash and will affect patterns of emersion and submersion, especially among regions with semi-diurnal, diurnal and semi-mixed tidal regimes, but incorporating that complexity is beyond the scope of this effort and does not change our conclusions or the implications of the concepts developed herein.

Wave height and tidal range data

Global maps of tidal range (Davies, 1980; Haslett, 2000) and satellite data on significant wave height which will be exceeded 50% of the time (Young & Holland, 1996) were used to generate Fig. 3A. For Fig. 3B, significant wave heights from the KNMI/ERA-40 Wave Atlas (Caires & Sterl, 2005a; Caires & Sterl, 2005b; Sterl & Caires, 2005) and tidal range from the TOPEX/POSEIDON 6.2 (TPXO6.2) data sets (Egbert, Bennett & Foreman, 1994; Egbert & Erofeeva, 2002) were used. The KNMI/ERA-40 Wave Atlas data were derived from the reanalysis of oceanographic and atmospheric data with the European Centre of Medium-Range Weather Forecast’s (ECMWF) Integrated Forecasting System coupled to the third generation wave forecast WAM model (Janssen et al., 2002; Komen et al., 1994). The diurnal tidal range was computed from the average maximum daily range for each day (MHHW – MLLW) using 10 available tidal constituents from TPXO6.2 (Egbert, Bennett & Foreman, 1994; Egbert & Erofeeva, 2002). Wave height to tidal range ratios were calculated and mapped for a 2° × 2° global grid (Fig. 3B) with Matlab 7.5 (MathWorks, Natick, Massachusetts) and ArcGIS 9.2 (ESRI, Redlands, California). Tide and wave heights can vary on small spatial scales; therefore, we expect there to be heterogeneity in the ratio of wave height to tidal range at smaller spatial scales than can be represented in a global map. For example, although the Hawaiʻian Archipelago is classified as wave-dominated, shorelines behind shallow reef crests which cause waves to break will be mostly tide-dominated. Microsoft Excel 2003 and Visual Basic for Applications (Microsoft Corp., Seattle, Washington) was the software modeling environment used to generate all statistics and figures unless otherwise noted above.

On a finer scale, we selected three specific sites that, on average, exhibit wave-dominated (Mokapu, Hawaiʻi), co-dominated (Humboldt, California), and tide-dominated conditions (Portland, Maine) to illustrate how their differences affect model predictions (locations marked with stars, Fig. 3A). Real, not predicted, historical data on significant wave height and tidal range (MHHW-MLLW) were extracted from the National Oceanographic and Atmospheric Administration (NOAA) National Data Buoy Center (www.ndbc.noaa.gov) and NOAA National Ocean Service oceanographic products (www.tidesandcurrents.noaa.gov), respectively. The wave buoys were located at Mokapu, Hawaiʻi (station 51202, 2000–2005), Humboldt South Spit, California (station 46212, 2004–2005), and Portland, Maine (station 44007, 1982–2001). The tide stations were located at Moku O Loʻe, Hawaiʻi (station 1612480, National Ocean Service Waimanalo tide correction applied), North Spit, California (station 9418767), and Portland, Maine (station 8418150).

Results

Derivation of shoreline benchmarks and zones

We begin by exploring the sinusoidal water level model which incorporates both tidal range and wave height for a single tidal cycle (Eq. (3), Fig. 1C). Four specific benchmarks associated with submersion and emersion can be derived from the basic model, Benchmarks 1–4 (Fig. 2). Benchmark one (B1) is the height of the upper reach of the wave crests at high tide and is defined as: (4) B1=AW+2AT+WTlow

where WTlow is the water level at low tide relative to MLLW (see Methods for definitions of other variables). Benchmark two (B2) is the height of the upper reach of wave crests at low tide and is defined as: (5) B2=AW+WTlow.

Benchmark three (B3) is the height of the lower reach of the wave troughs at high tide and is defined as: (6) B3=2AT+WTlow−AW.

Benchmark four (B4) is the height of the lower reach of wave troughs at low tide and is defined as: (7) B4=WTlow−AW.

Figure 1 Model of shoreline water level over a single tidal cycle.

Representation of water level incorporating (A) only tidal range, (B) only wave height, and (C) both tidal range and wave height. The tidal amplitude (AT), tidal period (PT), wave amplitude (AW), and wave period (PW) are noted.

Figure 2 Wave-tide model of shoreline water level.

Using three ratios of wave height to tidal range: 0.1:1 (A, D, G), 1:1 (B, E, H), and 2:1 (C, F, I), we show shoreline water level and shoreline benchmarks. The four shoreline benchmarks predicted by the model (B1–B4) are demarcated by colored lines and the zones they bracket are shown in panels D–F. Relative wave energy, continuous emersion time and submersion time are diagrammed in the conceptual models in panels G–I.

In the model, benchmarks one (blue) and four (black) mark the boundaries above which there is constant emersion and below which there is constant submersion, respectively (Fig. 2). Benchmark two (green) marks the lowest position on the shore that experiences tidally induced periods of emersion, and benchmark three (pink) marks the highest position on the shore that experiences tidally induced submersion. Benchmarks 1–4 can be used to define discrete shoreline zones. Benchmarks one (blue) and two (green) bracket a zone of tidally induced emersion (Figs. 2H and 2I), that we term the emergent tidal zone (vertical bars Figs. 2E and 2F). Similarly, benchmarks three (pink) and four (black) demarcate a zone of tidally induced submersion (Figs. 2H and 2I) that we label the submergent tidal zone (horizontal bars, Figs. 2E and 2F). Where these zones overlap (Figs. 2D and 2G), a zone of tidally induced emersion and submersion occurs – the traditional notion of an “intertidal zone” and roughly equivalent to the mid littoral zone of Stephenson & Stephenson (1972). A third zone, bracketed by benchmarks two (green) and three (pink), is sandwiched between the emergent and submergent tidal zones when wave height is greater than tidal range and is termed the wave zone because it is washed by waves at both high and low tide (Figs. 2F and 2I).

Derivation of physical categories for shorelines

Our model of shoreline water level predicts three primary categories for intertidal shores: tide-dominated, wave-dominated, and co-dominated (Fig. 2). When a shore is tide-dominated, tidal range is much greater than wave height and there is substantial overlap of the emergent and submergent tidal zones (Figs. 2A, 2D and 2G). When a shore is wave-dominated, wave height is much greater than tidal range and all three zones (emergent, wave, and submergent) exist independently (Figs. 2C, 2F and 2I). The third category, co-domination, occurs when wave height = tidal range, and is characterized by non-overlapping emergent and submergent intertidal zones, but no wave zone (Figs. 2B, 2E and 2H).

Global categorization of shorelines

A strength of our intertidal zonation model is that it can be applied across a variety of spatial and time scales. At a global spatial scale over a number of years, averaged data for wave height and tidal range can be used to classify shores as tide-, wave-, or co-dominated (Fig. 3). The most striking feature of Fig. 3 is that the majority of offshore oceanic islands are predicted to be wave-dominated (Fig. 3B) and the majority of continental shores are predicted to be tide-dominated on average (Fig. 3A). There are also numerous areas of co-domination and wave-domination on continental shores. Wave-dominated shores are most prevalent in the Southern Ocean where winds blow around the globe with few land barriers and in freshwater lakes which have negligible tidal action. It should be noted that tidal range and wave heights can vary on small spatial scales; therefore, we expect there to be heterogeneity in the ratio of wave height to tidal range at smaller scales than can be represented in an averaged global map (see the section below). This is not a failure of the model, rather it is an issue with the scale of the data loading the model. For example, although the Hawaiʻian Archipelago is classified as wave-dominated, shorelines behind shallow reef crests which cause waves to break will be mostly tide-dominated. If wave height data were collected on a fine scale to decipher fore reef from back reef, this pattern would be captured. Additionally, there is temporal variation in tidal range and wave heights which will cause temporal variation in the positions of Benchmarks 1–4, and could lead to temporal variation in the classification of points on the shore as tide-, co-, or wave dominated.

Figure 3 Global map classifying shorelines according to relative wave and tidal ranges.

(A) Shorelines classified as tide-dominated (mean diurnal tidal range ≫ mean significant wave height), co-dominated (mean diurnal tidal range ≈ mean significant wave height), and wave-dominated (mean significant wave height ≫ mean diurnal tidal range. (B) Ocean area color-coded by the ratio of wave height to tidal range. The locations of the sites featured in this study are indicated with stars – from left to right, Mokapu, Hawaiʻi; Humboldt, California; and Portland, Maine.

Temporal variation in zonation

It is important to recognize that there is a continuum from tide-, to co-, to wave-domination with no hard and fast boundaries, and the positions of Benchmarks 1–4 move as tidal range and wave height varies. Although the long term averages presented in Figs. 3 and 4A–4F are informative, tidal range and wave heights are highly variable in time. Consequently, it is important to be able to evaluate the variation in the position of Benchmarks 1–4 and the associated physical littoral zones through time. We applied Eqs. (4)–(7), for predicting the shore level of Benchmarks 1–4, to hourly wave height and tide measurements taken at Portland, Maine; Humboldt, California; and Mokapu, Hawaiʻi from June 2005–June 2006 to evaluate the variance in the predicted positions of the shoreline benchmarks on a fine temporal scale for one year (Figs. 4G–4I). The low tide line (WTlow) and tidal amplitude (AT) for Eqs. (4)–(7) were defined as the low tide level and the tidal amplitude during each semidiurnal tidal period, maximizing variation induced by tides.

Figure 4 Coastal water levels and physical zones across three representative sites.

Model of coastal water level (Eq. (3), panels A–C), derived benchmarks and physical zones (Eqs. (4)–(7), panels D–I), and histograms of the hourly frequency of the ratio of wave height to tidal range (panels J–L) applied to three representative wave-exposed sites along a continuum from tide- to wave-domination (Portland, ME; Humboldt, CA; and Mokapu, HI respectively). Panels (A)–(C) display the water level during one tidal cycle (12.2 h oscillation period) from MHHW to MLLW at the yearly mean significant wave height (shorter oscillation period) plotted against time. Panels (D)–(F) display the physical zones we derive from our model of coastal water level. Panels (G)–(I) show the hourly determination of Benchmark 1 (blue line), Benchmark 2 (green line), Benchmark 3 (fuscia line), and Benchmark 4 (black line), plotted against time from June 2005 to June 2006. Note that the rank positions of Benchmark 2 and Benchmark 3 switch from panels (A), (D) and (G) to panels (C), (F) and (I).

As expected, there was considerable fine scale variation in the model-predicted positions of the benchmarks at all three sites, which varied by up to 3.8 m, 4.4 m, and 2.0 m at Portland, Humboldt, and Mokapu, respectively (Figs. 4G–4I). Portland is classified as tide-dominated (ratio of wave height to tidal range <0.5) for 82% of the year-long period (Fig. 4J, notice that B2 is consistently lower on the shore than B3 indicating the overlap of the emergent intertidal and submergent tidal). Mokapu is classified as wave-dominated (ratio of wave height to tidal range >1.5) for 99% of the year-long period (Fig. 4L, notice that B2 is consistently higher on the shore than B3).

Humboldt is an interesting case because it is predicted to be co-dominated for 48% and wave-dominated for 48% of the year-long period on a fine temporal scale (Fig. 4K), but was classified as co-dominated based on the yearly averaged diurnal tidal range and yearly mean significant wave height (Fig. 4B). While there were portions of the winter where the significant wave heights were consistently at least 1.5 times greater than the diurnal tidal range due to large waves, most wave-dominated conditions were caused by large differences in the daily semidiurnal tidal ranges - the smaller tidal range being wave-dominated and the greater tidal range being co-dominated (causing the benchmark lines to appear thick in Fig. 4H). Ultimately, Humboldt tends to be co-dominated by waves and tides on a diurnal scale in the summer and can be wave-dominated in the winter, but also tends to alternate between co-domination and wave-domination on a fine semidiurnal temporal scale. Overall, our model of coastal water level can be used to effectively illustrate how patterns of wave and tide induced immersion vary at the scale of a single tidal period and hundreds of tidal periods, facilitating the physical description of sites in a concise and informative manner.

Discussion

We sought to develop a model of coastal water level that incorporates tidal range as well as wave height to better understand patterns of immersion and emersion on shorelines. The wave-tide model of shoreline water level, composed of two sinusoidal signals (Fig. 1), is simple, highly scalable, and can be used to predict fairly complex patterns in coastal water level. The model predicts the existence of four discrete shoreline benchmarks based upon patterns of submersion and emersion (Figs. 2A–2C). These four benchmarks bracket up to three primary physical zones on littoral shores (Fig. 2F): (1) the zone that is continuously emergent at low tide (emergent tidal zone), (2) the zone that is continuously submerged at high tide (submergent tidal zone), and (3) the zone that is continuously washed by waves regardless of tidal level (wave zone). The magnitude of wave height relative to tidal range controls the existence of these zones, and can be used to categorize shores as tide-, co-, or wave-dominated (Fig. 2).

The most familiar condition is that of tide-domination, where a large portion of the shore is characterized by tidally induced periods of both emersion and submersion – characteristics that are accurately predicted by our model with an overlap of the emergent and submergent tidal zones (Figs. 2A and 2D). For the most part, the world’s shores have been perceived to be subject to these tide-dominated conditions with only secondary modifications from wave action and other factors (Ricketts et al., 1985). The wave-tide model, however, predicts large portions of the earth’s shores are co-dominated and even wave-dominated (Fig. 3), where tidally induced emersion and submersion are spatially disconnected on the high and low shore, respectively (Fig. 2). Under co- and wave-dominated conditions, no position on the shore is predicted to experience periods of both emersion and submersion on a tidal schedule, unlike the classic “intertidal” conceptual model. Positions higher on the shore are alternately emerged and washed by waves due to tidal action (emergent tidal zone) while the low shore is alternately submerged and washed by waves (submergent tidal zone) (Fig. 2). During wave-dominated conditions, the mid shore is perpetually washed by waves regardless of tide (wave zone), thus waves modulate the vertical positions of emersion and submersion on shorelines. Importantly, because tidal range is variable and wave height is highly variable, iteration of the model over time and space predicts that the size and position of these zones will be highly variable (Fig. 4).

Fine-scale spatial variation

We have demonstrated the application of our model to large temporal and spatial scales, as well as fine temporal scales and midrange spatial scales, but it is also worthwhile to note that our shoreline zonation model can be applied at fine spatial scales of cm to m. Waves and more importantly, wave run-up can be influenced by many factors such as bathymetry, shoreline topography, slope, aspect, porosity, and rugosity. The model’s wave height parameter could be replaced with wave run-up parameters and it will reflect differences on fine spatial scales. Varying levels of wave exposure will affect the zonation predicted by the model. For example, shallow reefs can cause waves to break, severely reducing wave height and run-up at the shoreline, relative to that recorded at a wave buoy or coarse satellite imagery. Even within a shore that has fairly consistent wave exposure, there are areas where waves run up higher than others. If a shoreline were outfitted with submersion sensors, the positions of the benchmarks and zones could be estimated. Indeed, fine scale analysis requires fine scale data.

Behavior of zones with varying wave exposure and tidal range

Increasing wave exposure is believed to simply elevate and expand the tidally defined zones on the shore (Lewis, 1964; Ricketts et al., 1985; Stephenson & Stephenson, 1972) – a concept that has remained unchallenged in the literature. However, this behavior is not predicted by the model we present. The position of the emergent tidal zone (vertical bars, Figs. 2D–2F) rises but that of the submergent tidal zone (horizontal bars, Figs. 2D–2F) sinks on the shore as wave height increases. The wave zone (wavy horizontal bars, Fig. 2F) expands, but the emergent and submergent tidal zones remain the same size as wave height increases. Therefore, increasing wave height while holding tidal range constant displaces the tidal zones both higher and lower on the shore; this leads to the contraction of the region of overlap between the two tidal zones, as well as the emergence and expansion of the wave zone (see progression from Fig. 2D to Fig. 2F). Increasing tidal range, while holding wave height constant, increases the size of emergent and submergent tidal zones which leads to the contraction of the wave zone, and the expansion of the emergent-submergent overlap zone. Thus, the effect of tidal range and wave height fluctuations over time on physical shore zones is predicted to be different than previously envisioned.

Physical properties and biological implications of the wave:tide zonation model

At a gross level, the model-predicted zones and their unique physical characteristics can be used to further predict likely sources of stress and disturbance on biological communities. (Figs. 2G–2I, Table 1). The pattern and duration of immersion and emersion can have major effects on the foraging of predators and physiological stresses such as temperature, desiccation, and irradiation (Menge & Olson, 1990). As the physical zones move due to temporal variability in wave height and tidal range, conditions are predicted to change, thereby influencing resident shoreline organisms.

Table 1 Predominant physical conditions and their corresponding biological consequences for each of the zones predicted by the wave:tide zonation model.

Characterizations of predominant physical conditions and their biological consequences based on the assumption that the shores are equally wave-exposed. The predominant physical conditions and their corresponding biological consequences are ranked with zero to four “X” to indicate the magnitude of effect for each category column.

Habitat	Wave:tide regime	Physical conditions	Biological consequences	
	Tide
dominated	Co-
dominated	Wave
dominated	Tidal
emersion	Tidal
submersion	Wave
wash	Wave
force	Relative
wave
energy	Physiological
stress	Physical
stress	Benthic
predation
efficiency	Pelagic
predation
efficiency	Terrestrial
predation
efficiency	
Emergent tidal zone		Present	Present	XXXX		XX	XXXX	XX	XXXX	XX	XX		XXXX	
Wave zone			Present			XXXX	XXXX	XXXX		XXXX	X	X	X	
Submergent tidal zone		Present	Present		XXXX	XX	XXXX	XX		XX	XXXX	XXXX		
Emergent-submergent overlap zone	Present			XXXX	XXXX	X	XXXX	X	XXXX	X	XXXX	XXXX	XXXX	

Emergent tidal zone- The emergent tidal zone is characterized by tidally-influenced periods of wave-wash and exposure to air (Fig. 2). Therefore, the substratum temperature will be alternately affected by the water, atmosphere and solar irradiation on a tidally defined cycle. The substratum temperature amplitude can be much greater in this zone than the other littoral zones we have defined. The level of drying can be great during low tides, depending on wave splash and humidity, and the level of hydrodynamic force will oscillate from substantial to negligible on a tidally defined cycle. Consequently, the biota residing in the emergent tidal zone will likely contend with the physiological stresses of temperature and desiccation when exposed to the atmosphere and the physical stress of wave force when wetted by the ocean. Terrestrial predators, such as birds, will have access to the emergent tidal zone as the tide recedes, but non-benthic aquatic predators, such as fish, will have limited access to this zone because it will be constantly washed by waves at high water (see Garrity, 1984; Garrity, Levings & Caffey, 1986). Benthic marine predators will have to contend with physiological stresses when exposed to the atmosphere and physical stresses when exposed to the aquatic sphere, potentially limiting their efficiency (sensu Menge & Sutherland, 1976; Menge & Sutherland, 1987).

Wave zone- Physiological stresses, such as temperature and desiccation, are minimal in the wave zone because it is constantly washed by waves, and the temperature will be largely affected by the ocean temperature, muting temperature amplitude relative to the emergent tidal zone (Fig. 2). On the other hand, physical stress, in the form of hydrodynamic force caused by wave action, will be constant and the wave zone will receive more wave energy than the emergent or submergent tidal zones. As a result of these conditions, terrestrial and aquatic predators will have to contend with constant wave wash to safely access the wave zone. It is likely that predation will be severely reduced in the wave zone relative to any other littoral zone (see Garrity, 1984; Garrity, Levings & Caffey, 1986).

Submergent tidal zone- Heat and desiccation stresses are negligible in the submergent tidal zone because it is either washed by waves at low tide or completely submerged at high tide, but never fully emergent (Fig. 2). Physical stress will be magnified during periods of wave wash. Terrestrial predators will have little access to the submergent tidal zone, without specific adaptations, but aquatic predators will have easy access during high tides.

Overlapping of the emergent and submergent tidal zones- These descriptions of the emergent tidal, submergent tidal, and wave zones only apply to co- and wave-dominated conditions. During tide-dominated conditions (Figs. 2A, 2D and 2G) the emergent and submergent tidal zones overlap (overlapping vertical and horizontal bars, Fig. 2D). The result of the overlap between the zones is that the area is characterized by discrete tidally defined periods of exposure to the atmosphere, wave wash, and submergence – the conditions traditionally ascribed to a classic “intertidal” zone (Fig. 2G). Biota residing in this area of overlap are subject to physiological stresses (temperature, desiccation, and irradiation) during low water, are submerged at high water, and are exposed to physical stress in the wave wash at some point between high and low water. Terrestrial and aquatic predators have windows of foraging opportunity at low and high water, respectively, but are restricted during the periodic wave wash events with the tidal rise and fall (Garrity, 1984; Garrity, Levings & Caffey, 1986).

The non-overlapping regions of the emergent tidal (vertical bars, Fig. 2D) and submergent tidal zones (horizontal bars, Fig. 2D) share properties with those zones under co- and wave-dominated conditions, but the periods of exposure to the atmosphere and submergence in the emergent and submergent tidal zones, respectively, are long relative to the periods of wave wash. Thus, physical stress caused by waves will be reduced to shorter periods of time in both the emergent and submergent tidal zones under tide-dominated conditions. Physiological stress is potentially great and predation by swimming aquatic predators should be limited in the non-overlapping portion of the emergent tidal zone. In contrast, physiological stress will be minimal and access by aquatic predators will only be limited only during short periods of wave wash in the non-overlapping portion of the submergent tidal zone.

Consequences of variability in tidal range and wave height

While the general qualitative conditions we attributed to the physical zones on littoral shores are predicted to remain relatively constant over time, significant variation in both tidal range and wave height is expected within sites as well as among them. This variability within sites can lead to variation in the position and existence of these zones on a variety of time scales (Figs. 2G–2I). Against this backdrop of long-term average conditions, any given position on a shore can be characterized by a temporary profile of another of the physical zones we have described. Given that stresses and disturbances are often the result of uncommon events, rare departures from average physical conditions are likely to be events of extreme stress and disturbance. For example, ∼0.5 m above MLLW at Mokapu, HI is almost always predicted to be in the wave zone (below B2 and above B3, Fig. 2I), but is occasionally in the lower emergent tidal zone. On these rare occasions where 0.5 m is in the emergent tidal zone, physical stress due to constant wave wash will be alleviated but physiological stress due to temperature, desiccation, and solar radiation, as well as foraging by terrestrial predators are much more likely to impact the biota.

In Portland, ME, ∼1.5 m above MLLW is predicted to be typically in the overlap of the emergent and submergent tidal zones, but in approximately 23 discrete high-wave events it is solely in the upper submergent tidal zone or the wave zone (Fig. 2G). During these rare events, physiological stress is expected to be greatly reduced, but physical stress is expected to be elevated. Overall, both the average long-term conditions and the impact of rare events are expected to be important determinants of community structure. Our model aids in the identification of both the long term average conditions and these rare stress/disturbance events and facilitates the generation of specific predictions of how each will affect resident biota at specific positions on the shore.

When considering the additional complexities of the effects of bathymetry on wave height and the effects of shore topography, slope, porosity, effective fetch, and rugosity on wave run-up (Burrows, Harvey & Robb, 2008; Hughes, 2004; Thomas, 1986), it is easy to foresee that the model-predicted environmental conditions, based solely on coarse scale measurements of wave height and tidal range, may not exactly match the actual conditions at particular positions on the shore. The model is only as good as the data driving it, thus fine scale tests of the model must include fine scale wave run-up and tidal data (e.g., Harley & Helmuth, 2003; Burrows, Harvey & Robb, 2008). The power of the model lies in its scalability and its ability to identify transitions from tide- to co- to wave-domination, define typical conditions and significant departures from the norm, and make general predictions about the stresses affecting the biotic processes operating on any given shore.

Relevance of a physical zonation model

The use of physical zonation (“critical tide levels”, Colman, 1933; Doty, 1946) as a tool to better understand ecological processes on littoral shores has been refuted by Underwood (1978) who demonstrated there are no sharp changes in inundation patterns due to tidal ebb and flow, and thus, no zonation. Our model of shoreline water level and physical zonation is predicated on the smooth sinusoidal behavior of tidal ebb and flow, and is thus wholly consistent with the findings of Underwood (1978). Yet, we identify critical, physically-defined benchmarks on the shore when considering wave action as well as tidal fluctuation. We further agree with Underwood (1978) that particular species are not likely to be strictly limited within or by these zones. We propose, however, that the conditions associated with these zones and their positions (as determined by the ratio of wave height to tidal range) will have prominent effects on physical and physiological stress and disturbance experienced by shoreline populations. The differential stresses in each zone will impact prominent biotic interactions such as predation, competition, and facilitation (Bruno & Bertness, 2001; Menge & Sutherland, 1987). Consequently, our model of physical zonation is an effective tool that can inform and facilitate our understanding of physical processes operating on littoral shores, and is a fundamental advancement beyond the classic intertidal concept.

Conclusion

Our model of coastal water level indicates that waves do not simply expand and elevate physical littoral zones that exist in the absence of waves. Rather, waves interact with tides to create up to three distinct physical zones (emergent tidal zone, wave zone, submergent tidal zone) that are characterized by unique submersion-emersion and hydrodynamic characteristics. The differential properties of these zones and the variability in their existence, overlap, and positions leads to specific and falsifiable hypotheses regarding differential regimes of physical and physiological and stress and disturbance for organismal shoreline populations. What is more, our model of littoral zonation is completely consistent with the findings of previous studies that have argued against the existence of intertidal zonation (Underwood, 1978; Benedetti-Cecchi & Cinelli, 1997), and is not hampered by a complete dependence on tidal patterns, geographic location, or the biological definition of zones. Thus, this model provides a unifying framework to better understand the physical littoral habitat and biotic stress regimes on shores, whether they are temperate, tropical, marine, or lentic.

We thank K Bridges, D Duffy, L Freed, A Kay, H Zaleski, and anonymous reviewers for providing insightful comments and constructive criticism. This is HIMB contribution HIMB 1565, and SOEST 8988.

Additional Information and Declarations

Competing Interests

Author Contributions

Robert J. Toonen is an Academic Editor for PeerJ. None of the other co-authors have competing interests.

Christopher E. Bird conceived and designed the experiments, performed the experiments, analyzed the data, contributed reagents/materials/analysis tools, wrote the paper.

Erik C. Franklin conceived and designed the experiments, analyzed the data, contributed reagents/materials/analysis tools, wrote the paper.

Celia M. Smith conceived and designed the experiments, contributed reagents/materials/analysis tools.

Robert J. Toonen conceived and designed the experiments, contributed reagents/materials/analysis tools, wrote the paper.

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
