# Peer review of "Between tide and wave marks: a unifying model of physical zonation on littoral shores"

_PeerJ, doi:10.7717/peerj.154_

## Round 0.1 · original submission · Minor Revisions

First. I want to thank you for submitting to PeerJ and also to apologize for the too long review time. I had ten people decline my invitation to review. Probably a record, but does seem to be getting harder and harder to find reviewers.

Both reviewers were supportive and only asked for very minor revisions. Can you do this quickly and resubmit? When I get it back, I will not send it out again and will make a final decision within 24 hours.

·

Basic reporting

I agree with the authors that this study represents an advancement of our understanding of physical processes operating on littoral shores. The authors present and conceptual model and modelling framework that can be applied across a range or temporal and spatial scales. The background leading to this study is presented succinctly and demonstrates the foundations for this body of work.

Experimental design

No comments

Validity of the findings

See above

Additional comments

Paragraph starting line 52: The authors may wish to consider expanding the discussion re community composition and wave exposure eg see Burrows et al 2008 Wave exposure indices from digital coastlines and the prediction of rocky shore community structure. Marine Ecology Progress Series 353: 1-12 and Thomas 1986 A physically derived exposure index for marine shorelines. Ophelia 25:1-13.

Line 79-87: I suggest bringing in a sentance to highlight that tidal regimes are also differ regionally,eg semi-diurnal, diurnal or semi-mixed,

Line 112: I think you are referring to Fig 3a not Fig 1.

Line: 159: incorrect spelling “co-domintated”

Reviewer 2 ·

Basic reporting

This paper presents a novel model that explores the interaction between tidal and wave forces in generating intertidal zonation patterns. I have heard the senior author speak at national meetings about this model and seen a couple versions of this paper previously.
It should have been published years (6 or so) ago when it was first presented, but it unreasonably offended established rocky intertidal ecologists who were too set in their ways to accept fresh thinking about the drivers of intertidal zonation. Frankly, this should have been an ecology paper and in textbooks. It improves upon all the earlier models of Coleman, Doty and the Stephenson’s.

The disparity between the intertidal mafia and a bright ambitious ecologist working alone in Hawaii w/o contact with high profile intertidal ecologists was that most established intertidal ecologists worked in habitats with large tidal ranges, ranges greater than local wave amplitudes. Bird was working in Hawaii where the tidal amplitude far exceeded the tidal amplitude. He did outstanding community assembly experimental shoreline ecology on his system, but since he was working alone in a system with very different parameters his work met resistance from the intertidal establishment on all his experimental work. This model incorporating these disparities in a general model is an outstanding example of originality coming out of a novel system that can teach us a general lesson. That his work was blackballed by established intertidal ecologist is a sad statement about how established dogma and their protagonists can hold back advancement based on myopic ideas.

This paper is already in perfect condition after numerous submissions, improvements and rewrites. Unfortunately, Bird has moved on to molecular evolution work where he is excelling, leaving the pettiness of rocky intertidal ecologists behind. He could have been an exceptional experimental community ecologist with better mentoring and more open-minded rocky intertidal researchers.

Experimental design

N/A

Validity of the findings

N/A

---

## Round 0.2 · accepted · Accept

I think the paper is good shape for production.